# To explain or not to explain?—Artificial intelligence explainability in clinical decision support systems

Julia Amann[1,◎,*], Dennis Vetter[2,3,◎], Stig Nikolaj Blomberg[4], Helle Collatz Christensen[4], Megan Coffee[5], Sara Gerke[6], Thomas K. Gilbert[7], Thilo Hagendorff[8], Sune Holm[9], Michelle Livne[10], Andy Spezzatti[11], Inga Strümke[12,13], Roberto V. Zicari[14,15], Vince Istvan Madai[16,17,18,◎], on behalf of the Z-Inspection initiative

1 Health Ethics and Policy Lab, Department of Health Sciences and Technology, ETH Zurich, Zurich, Switzerland, 2 Frankfurt Big Data Lab, Goethe University Frankfurt am Main, Germany, 3 Computational Vision and Artificial Intelligence, Goethe University Frankfurt am Main, Germany, 4 University of Copenhagen, Copenhagen Emergency medical Services, Denmark, 5 Department of Medicine and Division of Infectious Diseases and Immunology, NYU Grossman School of Medicine, New York, United States of America, 6 Penn State Dickinson Law, Carlisle, PA, United States of America, 7 Digital Life Initiative, Cornell Tech, New York, NY, United States of America, 8 Cluster of Excellence "Machine Learning: New Perspectives for Science"—Ethics & Philosophy Lab University of Tuebingen, Germany, 9 Department of Food and Resource Economics, Faculty of Science University of Copenhagen, Denmark, 10 Google Health Research, London, United Kingdom, 11 Industrial Engineering & Operations Research Department, University of California, Berkeley, United States of America, 12 Department of Holistic Systems, Simula Metropolitan Center for Digital Engineering, Oslo, Norway, 13 Department of Engineering Cybernetics, Norwegian University of Science and Technology, Trondheim, Norway, 14 Yrkeshögskolan Arcada, Helsinki, Finland, 15 Data Science Graduate School, Seoul National University, Seoul, South Korea, 16 QUEST Center for Responsible Research, Berlin Institute of Health (BIH), Charité Universitätsmedizin Berlin, Germany, 17 CLAIM—Charité Lab for Artificial Intelligence in Medicine, Charité Universitätsmedizin Berlin, Germany, 18 School of Computing and Digital Technology, Faculty of Computing, Engineering and the Built Environment, Birmingham City University, United Kingdom

◎ These authors contributed equally to this work.
* julia.amann@hest.ethz.ch

**Data Availability Statement:** All data are in the manuscript and/or supporting information files.

**Funding:** JA was supported by funding from the European Union's Horizon 2020 research and

## Abstract

Explainability for artificial intelligence (AI) in medicine is a hotly debated topic. Our paper presents a review of the key arguments in favor and against explainability for AI-powered Clinical Decision Support System (CDSS) applied to a concrete use case, namely an AI-powered CDSS currently used in the emergency call setting to identify patients with life-threatening cardiac arrest. More specifically, we performed a normative analysis using socio-technical scenarios to provide a nuanced account of the role of explainability for CDSSs for the concrete use case, allowing for abstractions to a more general level. Our analysis focused on three layers: technical considerations, human factors, and the designated system role in decision-making. Our findings suggest that whether explainability can provide added value to CDSS depends on several key questions: technical feasibility, the level of validation in case of explainable algorithms, the characteristics of the context in which the system is implemented, the designated role in the decision-making process, and the key user group(s). Thus, each CDSS will require an individualized assessment of explainability needs and we provide an example of how such an assessment could look like in practice.

innovation program under grant agreement No. 777107 (PRECISE4Q). DV received funding from the European Union's Horizon 2020 Research and Innovation Program 'PERISCOPE: Pan European Response to the ImpactS of COvid-19 and future Pandemics and Epidemics' under grant agreement no. 101016233, H2020-SC1-PHE-CORONAVIRUS-2020-2-RTD and from the European Union's Connecting Europe Facility program 'xAIM: eXplainable Artificial Intelligence for healthcare Management' under grant agreement no. INEA/ CEF/ICT/A2020/2276680, 2020-EU-IA-0098. The funders had no role in study design, data collection and analysis, decision to publish, or preparation of the manuscript.

**Competing interests:** VIM reported receiving personal fees from ai4medicine outside the submitted work. There is no connection, commercial exploitation, transfer or association between the projects of ai4medicine and the results presented in this work.

## Introduction

Machine learning (ML) powered Artificial intelligence (AI) methods are increasingly applied in the form of Clinical Decision Support Systems (CDSSs) to assist healthcare professionals (HCPs) in predicting patient outcomes. These novel CDSSs have the capacity to propose recommendations based on a plethora of patient data at a much greater speed than HCPs [1]. In doing so, they have the potential to pave the way for personalized treatments, improved patient outcomes, and reduced health care costs.

In laboratory settings, proof-of-concept AI-based CDSSs show promising performance [2,3]. In practice, however, AI-based CDSSs often yield limited improvements [1,4–9]. A possible explanation for this might be that in cases where the AI system's suggested course of action deviates from established clinical guidelines or medical intuition, it can be difficult to convince HCPs to consider the systems' recommendations rather than dismissing them a priori.

It is important to note that many AI algorithms, especially the popular artificial neural networks (ANN), are so-called "black boxes," because the inner workings of the algorithm remain opaque to the user. This situation can lead to a lack of trust in the black-box AI system, which is an important barrier to CDSS adoption [1]. Consequently, finding ways to foster HCPs' trust in CDSSs is critical to enable their wide-range adoption in clinical practice [1,10,11].

An often referenced way to foster trust is to increase the system's transparency [12], and an important part of increasing transparency is the application of explainability [13]. While a multitude of different approaches exists to explain the inner workings of an AI system, each of these approaches entails certain advantages and challenges [14–17]. In addition, what constitutes a "good" explanation depends on multiple factors, such as the target audience and intended use case [18–24]. In fact, it even remains contested whether explainability should at all be a requirement for CDSSs. While the majority of the academic community seems to lean towards explainability as instrumental, desirable, and potentially necessary [11,14,25–27]. There are also compelling arguments against this predominant view [28–30].

According to the European Commission's High-Level Expert Group on AI (AI HLEG), transparency is one of the key requirements for trustworthy AI. Explainability is but one measure of achieving transparency; other measures include, for example, detailed documentation of the used datasets and algorithms, as well as an open communication on the system's capabilities and limitations. While AI HLEG deems explainability useful, the expert group does not consider it necessary for trustworthy AI systems. However, according to AI HLEG, in systems without explainability, efforts need to be made to include other measures of transparency [13].

The present paper contributes to this discourse by reviewing some of the key arguments in favor and against explainability for CDSSs by applying them to a practical use case. Specifically, the paper builds on and extends the trustworthy AI assessment of an AI system deployed and used to detect cardiac arrest in an emergency call setting in Denmark (Z-Inspection process®) [31]. By exploring different development scenarios for the system—with and without explainability—this paper provides a nuanced account of the role of explainability for CDSSs that goes beyond purely theoretical considerations.

## Terminology

A considerable challenge in the field of explainable AI is the lack of a commonly accepted interdisciplinary terminology. It is possible—as we have done so far—to refer to explainability as an overarching, general concept; this is also how the AI HLEG uses the term [13]. Using this terminology, it can be defined as additional information—next to the performance of the AI system—on *how* an AI system arrives at a certain output. General explainability can have

different scopes: *global* explainability provides information on the model as a whole, while *local* explainability provides information on a specific prediction.

When referring to different algorithms that can achieve explainability, there are unfortunately a multitude of different terms in the field that are frequently used interchangeably and without a fixed definition, especially across disciplines [22,25,28]. Thus, there is a need to define the usage of these terms for the current work. In the following, we will distinguish between interpretable and explainable algorithms. We find the following definition useful for our use case: While both explainable and interpretable algorithms have the same goal in common as defined by the previous overarching general definition, they are distinct with regard to the technology applied. In this context, interpretable algorithms encompass all methods where the model's decision process can be inherently and intuitively understood by the intended user. Common examples are linear and logistic regression techniques, where the importance of features can be inferred from the feature weights. Another example is decision trees which humans can intuitively interpret. In contrast, explainable algorithms refer to methods that try to open black-box models post hoc. These methods are often themselves interpretable algorithmic models that approximate the black-box models. An explanation for a black-box model can then, for example, be obtained through an interpretable approximation (see, e.g., SHAP [32] or LIME [33]).

Since a main concern of this work is the practical implications of the choice of explainability, we will build on this introduced distinction. Although there is some debate about the terminology of the terms, when referring to the general concept of explaining machine learning models, we will call it *explainability*. This can be achieved either by using inherently *interpretable algorithms* (which provide *interpretations*) or by using *a black-box algorithm* and an additional *explanation algorithm* (which provides *explanations*). These *interpretations* or *explanations* are what the user is interacting with (Fig 1).

## Brief review of explainability for CDSSs

### Explainability: Opportunities and merits

Explainability methods hold great potential for CDSSs. Explaining the inner workings of AI systems can facilitate finding potential flaws and identify the root causes of errors more efficiently [34]. On the other hand, explainability allows HCPs to judge whether the system's output is trustworthy [35]. Moreover, it may help them to better communicate with patients why a specific course of action was recommended by the AI system and to promote trust in the HCP-patient-relationship [10,13,18,35]. Explainability also enables users to verify that the system does not rely on artifacts or noise in the training data, and can be used to assess whether the system is fair in its decisions, particularly when training data may include a biased or incomplete picture of the population [36]. Furthermore, explanations have the potential to reveal new insights regarding what the AI system learned from data, and better understand what the algorithm optimized for and the related trade-offs [37]. It has also been shown that providing HCPs with an AI system's qualified second opinion can increase diagnostic accuracy over that of either the AI system or the HCP alone [38]. Some authors even argue that explainability constitutes a normative requirement for medical applications [18].

### Explainability: Challenges and drawbacks

Especially in the medical domain, explainability might be misleading and, in fact, not always necessary.

Arguing against the general concept of explainability, it was posited that HCPs prefer effective use of information from trustworthy sources over a complete understanding of how the

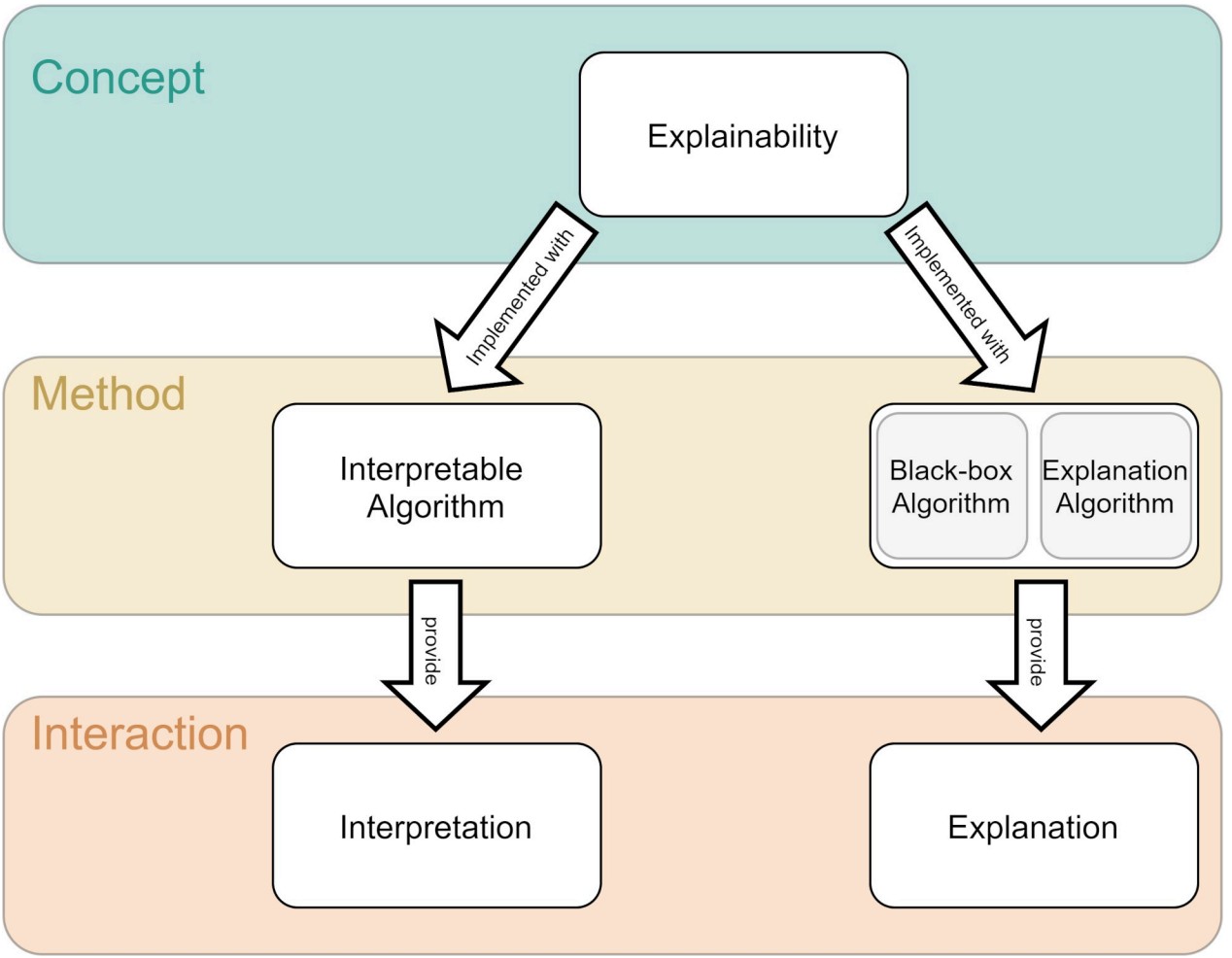

**Fig 1. Terminology.** Given that there is no commonly accepted terminology, we defined the following terms for this work: When referring to the general concept of explaining machine learning models, we will call it explainability. This can be achieved either by using inherently interpretable algorithms (which provide interpretations) or by using a black-box algorithm and an additional explanation algorithm (which provides explanations). These interpretations or explanations are what the user is interacting with.

information was generated [24,29]. Here, the actual usability and performance of a system are considered more important than the system's ability to explain its outputs, as long as these outputs are shown to be sufficiently accurate and validated [24,39,40]. Experimental evidence also suggests that explainability might not actually make users more likely to follow an AI system's predictions and even make it more difficult to recognize wrong predictions [36]. An additional argument against explainability is that, in some cases, a universal ground truth that HCPs agree on might not exist [24,41]. Such cases pose a high, and perhaps even unfeasible, demand on the level of detail of the explanations.

There is also a risk associated with users misunderstanding plausible associations detected by AI systems. A key point here is that while an AI-based CDSS may be able to provide explanations for its predictions, clinicians might wrongfully assume causality. However, given that explanations are correlation-based, they are susceptible to error due to random factors, like overfitting and spurious correlation, and thus require clinical inference to determine causal reasons [29]. Contrary to expectations, using an easy-to-understand and transparent model can decrease the likelihood of users detecting errors in the model, as opposed to using tools

without explainability. A possible reason for this is that the user then needs to process too much information to focus on error detection [36]. It is also possible to craft misleading explanations that look trustworthy to the user and that, while they technically reflect the model's decision process with high accuracy, omit parts of the explanation and ultimately decrease the level of trust in the model [42–44] by allowing vendors to use explainability to "fairwash" their AI systems [22].

An additional technical challenge is that state-of-the-art deep learning models are not inherently interpretable [18,28,45]. Existing methods aimed at generating explanations for these deep learning models often rely on approximations from explanation algorithms. This leads to a real danger that these approximations—and therefore the explanation they provide—do not accurately represent the model for some inputs. Another possible reason for an inaccurate representation of the original model is that there is no guarantee that the approximation-based explanations use the same features as the original model. The real possibility of inaccurate explanations can make it harder for clinicians to trust the explanations, and consequently also the model they are explaining—contrary to the intended purpose of these explanations [45–47].

Furthermore, what constitutes a good explanation depends on who is interacting with it [18,19,21–24,37] and there is no quantitative way of determining the most useful interaction in advance [19,36,48,49].

Due to these limitations in current techniques for generating explanations, some experts suggest the use of inherently transparent models such as regression, decision trees, or rule lists [45,50]. While the decision process of these models is more interpretable for end-users than that of, for example, a deep neural network (DNN), these models can only be considered interpretable as long as they follow certain limitations, such as with respect to model complexity or number and type of input features, which begs the question of whether inherently transparent models actually exist [28,39,51].

Others propose to use counterfactual explanations [47,52–54], which are explanations of the form "if the input were this new input instead, the system would have made a different decision" [22]. These explanations resemble the often contrastive way of human explanations [21] and are therefore easy to understand. However, in practice, it can be difficult and computationally expensive to find meaningful and useful counterfactual examples [53–55].

## Methodology

There is considerable uncertainty about the expected utility and appropriate implementation of explainability in AI-based CDSSs. To advance current theoretical considerations on the topic, we performed a normative analysis using socio-technical scenarios to provide a nuanced account of the role of explainability for CDSSs in a concrete use case. More specifically, we adopted a bottom-up approach where our analysis was informed by a real-world application with the aim to make abstractions for AI-powered CDSS, more broadly. We focused our analysis on three layers: technical considerations, human factors, and the designated system role in decision-making.

The case presented here draws on four consecutive workshops (N = 23, N = 29, N = 23, N = 8) which involved the initiators of the technology (coauthors S.N.B., H.C.C.) and members of the Z-Inspection® initiative, representing researchers and practitioners from medicine, computer science, and social sciences, including the authors of this paper [56].

Workshops were held online and recorded using the video conferencing software Zoom. The aims of the workshops were to gain a deeper understanding of the functioning and practical implementation of the AI system, assess current evidence, and engage in an informed

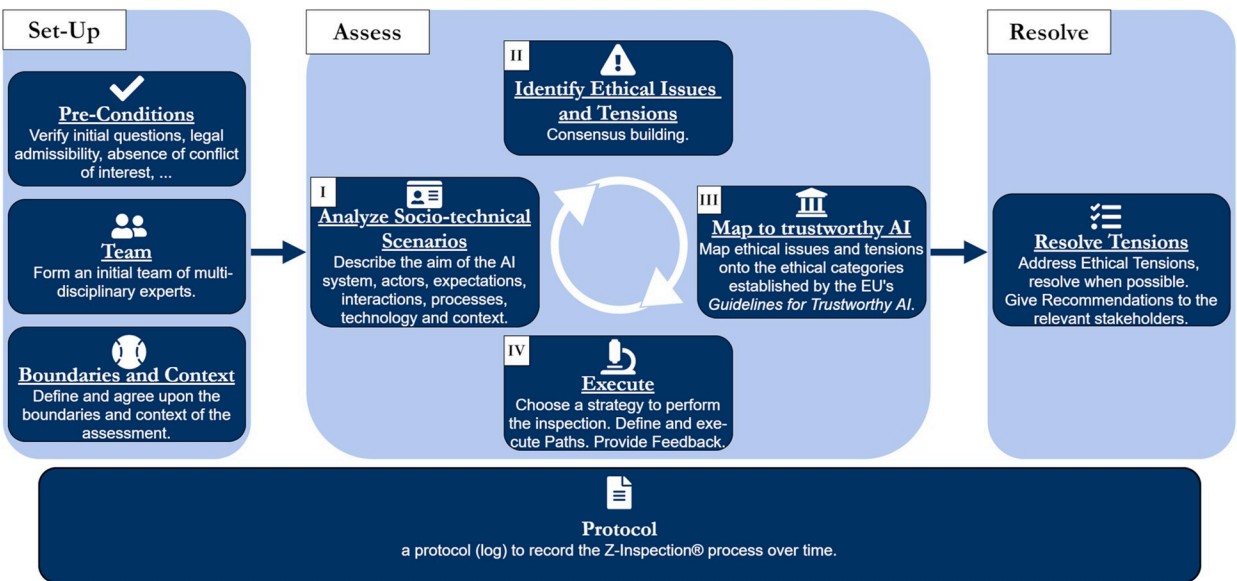

**Fig 2. Z-Inspection® process.** This figure has been reproduced from Zicari RV, Brodersen J, Brusseau J, Düdder B, Eichhorn T, Ivanov T, et al. Z-Inspection®: A Process to Assess Trustworthy AI. IEEE Trans Technol Soc. 2021 Jun;2(2):83–97. [56]

discussion about the technical, ethical, and legal implications of the system. The chat protocol and transcripts of the workshop, as well as relevant publications [57,58] made available by the initiators of the technology (coauthors S.N.B., H.C.C.), served as the basis for an interdisciplinary assessment of system trustworthiness following the Z-Inspection® process [31,56], which is guided by the "Ethics Guidelines for Trustworthy AI" by the European Commission's AI HLEG [13]. The report of the assessment has been published elsewhere [31]. Fig 2 presents an overview of the assessment process.

The present paper draws on this assessment to inform the development of two scenarios, which explore different pathways for the particular use case focusing on "Transparency", one of the key requirements for trustworthy AI [13]: the AI system as a black box (scenario 1) and as an explainable model (scenario 2). Specifically, the initial assessment of the system and consequent mapping of ethical issues identified lack of explainability as an ethical issue and possible explanation for the lack of dispatchers' compliance with the system's recommendations [31] (Fig 3). In devising the scenarios presented here, we further relied on prior empirical, conceptual, and normative work on the concept of explainability in CDSSs. We aimed to capture the various disciplinary perspectives on explainability, ranging from philosophy, to medicine, to computer science, and law [14,17,18,59,60]. Against this evidence base, we assessed the role of explainability in clinical decision support systems, taking the case of an AI-based CDSS used to recognize out-of-hospital cardiac arrest as a case in point.

## Use case

The AI-based CDSS in use recognizes out-of-hospital cardiac arrest (OHCA) from audio files of calls to the emergency medical dispatch center. OHCA is a medical condition where the heart stops beating due to a failure in signal transmission within the heart [61]. This means that at the time of the call, the patient is clinically dead, and therefore the caller is never the patient but a bystander. If OHCA is detected, the dispatcher instructs the caller to perform chest compressions (CPR), as every minute without CPR reduces the chances for successful resuscitation and drastically increases the risk for permanent complications, like brain

## ID Ethical Issue: E6, The AI tool is not Interpretable

**Description**

The system outputs cannot be interpreted, leading to challenges when dispatcher and tool are in disagreement.

**Map to Ethical Pillars/Requirements/Sub-Requirements (Closed Vocabulary)**

Explicability > Transparency > Explainability.

**Narrative Response**

The tool lacks explainability, which might lead to several challenges. First, outcomes are based on a transcription of the conversation between dispatcher and caller. It is not clear what is used from these transcripts to trigger an alert. This lack of transparency may have contributed to the noted lack of trust among the dispatchers, as well as the limited training of the users. Second, there is a lack of transparency regarding whether and which value judgments went into the design of the model. Such value judgments are important because explaining the output is partly a matter of accounting for the design decisions that humans have made.

## ID Ethical Tension (Open Vocabulary): ET6

Kind of tension: True dilemma.

Trade-off: *Accuracy vs. Explainability.*

Description: The tool lacks explainability but explainable AI systems may be less accurate than non-interpretable models.

## ID Ethical Tension (Open Vocabulary): ET7

Kind of tension: True dilemma.

Trade-off: *Security vs. Accessibility.*

Description: The system should be transparent and available to various stakeholders, but also must have safeguards to resist external threats that may limit transparency conditions.

**Fig 3. Ethical issue identified during the initial assessment of the use case [31].**

damage. Survival is unlikely after 15 minutes without CPR [61,62]. This time sensitivity makes the detection of OHCA one of the key quality indicators for emergency lines across Europe [60].

In Copenhagen, OHCA related emergency calls are rare, comprising less than 1% of all calls to the 112 emergency line [58]. Dispatchers are able to correctly identify about 75% of all OHCA cases. To improve these rates, an AI system was designed with the goal of supporting the dispatchers in reacting faster and more reliably to the time-sensitive OHCA emergencies by learning new and effective indicators for OHCA from a large number of previous calls.

During calls to the 112 emergency line, the system analyzes the dispatcher's questions and the caller's answers in order to determine whether an OHCA event is happening. If it detects such an event, it displays an alert to the dispatcher. Protocol dictates that the dispatcher follows up on alerts and re-addresses with the caller if the patient is conscious and breathing to establish whether the patient is in cardiac arrest. From that point on it is the dispatcher's decision if they want to follow up further or instruct the caller to start CPR while the ambulance is on the way. After these follow-up questions, the dispatcher is also free to disregard the system's alert if they suspect a false alarm. At no point does the AI system suggest questions or perform other actions to steer the conversation. The system is only a support for the dispatcher who is in charge and responsible at all times.

A retrospective study showed that the system alone was able to identify cases of OHCA with a higher sensitivity and faster than the dispatchers. In some cases, the system was even able to detect the OHCA minutes before the dispatcher [57]. A randomized clinical trial confirmed the system's performance but also found no significant differences between the sensitivity of dispatchers supported by the CDSS and those not supported by the CDSS. It showed that the dispatchers rarely adequately acted on the system's alert, despite being aware of its demonstrated performance [58]. During the assessment, a lack of trust was identified as a possible cause [31]. One reason for this lack of trust might be the high number of false positives produced by the system. In addition, the system did not provide any explanations regarding which part(s) of the conversation triggered the alert or its level of confidence. Consequently, the dispatcher could not easily judge whether the system's output was appropriate in a given case and, therefore, likely chose to disregard the system and trust their own judgment instead.

## Results

In the following, we present the two socio-technical scenarios that outline the implications the foregoing or addition of explainability would have for the use case at hand and what measures could be adopted, respectively, to increase the dispatchers' trust in the system.

### Scenario 1—Foregoing explainability

In the first scenario, we outline the potential further validation and development process for the status quo, i.e., the cardiac arrest prediction tool as a black box (without implemented explainability). As described above, the current status is an AI tool where exploratory research indicates higher sensitivity and lower time to recognition of the system, but where the dispatchers did not demonstrate absolute compliance to the tool in a first randomized clinical trial, i.e., they rarely adequately acted on the system's alerts.

**Technical considerations.** The first layer to explore relates to the technical aspects of omitting explainability. Without explainability, the sole criterion to translate this system into clinical practice and to justify widespread clinical use is an excellent performance evidenced by a robust, high-quality clinical validation through randomized controlled trials [39,40,63]. For this purpose, major emphasis needs to be put on maximizing the technical excellence of the system. Additionally, a clinical validation strategy must be developed. Because black box techniques are sensitive to the datasets used and the diversity of training data, which may include limitations users of the tool may not be aware of, this strategy needs to ensure that excellent performance is generalizable to the full range of use-case variability, including population characteristics and other relevant factors (e.g., time of calls or technical settings of the callers). In the current case, a reasonable first target would be the country of Denmark. Thus, prior to widespread testing, it needs to be ensured that the system can deal with all potential dialects and accents present in the Danish population, including those of minorities and marginalized groups, to demonstrate generalizability. This is crucial in the case of no explainability since the dispatcher has no means to identify a wrong prediction when it occurs, and deep learning systems can make predictions with high confidence even when these are faulty [64,65]. If sufficient representation of minorities cannot be guaranteed, the system should at least detect when it encounters a not sufficiently understood dialect [66,67] to then notify the dispatchers whilst also refraining from making a prediction.

**Human factors.** The second layer of our analysis relates to the human factors that must be considered when omitting explainability. A general challenge for clinical validations of CDSSs without explainability is that the user who needs to be convinced to trust the system is at the same time the user operating the system in a prior clinical validation. If—due to

whichever reason—the user does not operate the system with confidence during clinical validation, the validation will fail. This is exactly what was observed in the randomized clinical trial of the system at hand. Thus, systems without explainability face the challenge that users in clinical validation studies must be convinced to trust the systems *without* sufficient evidence of clinical efficacy. In such cases, a common proposition for fostering the users' trust in the system is to include them more actively in the design and evaluation process [13,19,24,35]. This inclusion could come, for example, in the form of a dedicated teaching module that contains a collection of all pre-validation evidence, put together in a way that it is understandable for the users, who are lay people with regards to AI technology. It should encompass how the AI system works, how generalizability was ensured in the training procedure, what the achieved performance measures mean in practice, and recommendations for interactions between the CDSS and its users, such as how to practically deal with the system's recommendations. This teaching module should be constructed during the design phase of the study to allow for comments and criticism from the user in the study to be taken into account.

**Designated role in decision-making.**   The third layer is the intended role of the system in the decision-making process. As previously mentioned, without explainability or other confidence estimates, there is no safeguard against confident false positives or negatives. This means that every prediction is presented to the user in a binary fashion i.e., alert or no alert, with the same confidence. In some use cases—including the present one—wrong predictions can have fatal consequences, i.e., a missed cardiac arrest. While it is true that the system is only intended for decision support, it can only have an impact if the dispatchers actually trust the system and align their action with the output of the system. The challenge remains, however, that the dispatchers will have no additional tools available to decide when to defer to the model. This challenge is intensified by the time-critical context of the emergency setting. It is, in fact, reasonable to say that the omission of explainability in any system similar to the one in the present use case alters the function of the system: It turns the decision-making process from one that was meant to be supported into one that is driven by the algorithm, essentially requiring the dispatchers to give up a considerable amount of autonomy in the decision-making process. And, depending on the use case and the psychological characteristics of the user group, the users might be more or less willing to defer to an automated system and to give up autonomy in the decision-making process, especially considering that they are still held responsible for the final decision.

## Scenario 2—Adding explainability

In contrast to scenario 1, we outline here the potential further development and validation process for the hypothetical scenario of adding explainability to the CDSS presented in our use case. The underlying assumption is that the dispatchers can better interact with a system if they know how or why a particular decision was made. This way, the CDSS can fulfill the role of an additional digital dispatcher, providing not only a second opinion but also additional insights into what can (in)form the decision. This helps the dispatchers weigh what the system has learned and determine whether the system's decision is applicable to the case at hand. This is especially relevant to the current case, as there is a potential to benefit from the system's higher sensitivity while at the same time using the dispatcher's expertise to mitigate the AI system's lower specificity.

**Technical considerations.**   When considering technical aspects of including explainability in the system, we first need to determine whether, and if so, how it might be achieved. Which type of explainability can be utilized depends on factors like data modality, the type of model, and the prediction task. In our use case, the recognition of OHCA is performed by DNNs [58],

which are generally considered black boxes because the computations they employ to map inputs to outputs are far too complex for humans to understand. However, generating justifications for predictions can be added to the objective function of a DNN, and additional explanation algorithms—such as LIME [33]—could be used to highlight the words or phrases from the previous conversation that influenced the CDSS's decision the most. The challenge here is that the model powering the CDSS is proprietary and therefore not easily accessible. While LIME is a general purpose explanation algorithm that can work with any black-box model, cooperation with the CDSS's vendor and access to the underlying model might enable more specialized explanation techniques [68–70]. Whether created with LIME or another method, explanations that highlight important pieces of the previous conversation could, for example, help the dispatcher see if a phrase they dismissed at one point became relevant later in the conversation, or if the system made its recommendation based on phrases the dispatcher does not consider important. Given the time-sensitive context, explanations would need to be intuitive and easy to grasp.

A consequence of adding explanations to the system is that the quality of the explanations and their usability (i.e., how they are presented to the dispatcher) should also be evaluated. First, it must be ensured that the approach used to derive explanation is methodologically sound. Despite numerous existing methods to derive explanations, many experts criticize the validity of these approaches, and there is no consensus regarding which explanation methods are superior in general. Hence, the explanations must be validated with respect to both their validity and their usefulness. This might require special training for the dispatchers on how to work with the explanations. It will also be important to involve dispatchers when assessing the explanations' validity. Including the dispatchers in design and validation is important because whether an explanation (or interpretation) is useful depends on the audience and context [18,19,21–24]. While there is some guidance on how to satisfy different explanatory needs [71] and attempts at metrics to compare generated explanations [72–74], there is no quantitative way to determine the best type of explanation for a given use case in advance [19,36,48,49].

**Human factors.**   Looking at human factors, it is reasonable to assume that explainability would facilitate validation and acceptance of this CDSS. Such information would need to be provided to dispatchers in an intuitive, easy-to-grasp format, for instance using visual cues that explain the system's recommendation. Under the assumption of technically accurate and useful explanations, it should then be easier for the user to interact with the system confidently. By providing explanations that are explicitly designed to be useful to the dispatchers, they will have more data available to make a decision on whether the system's decision (and the explanation) is accurate and applicable to the current use case. Assessing an explanation is also likely easier for the dispatchers, as both the explanations and dispatchers ultimately apply a similar process to derive a decision, and therefore the dispatcher can better apply their experience and intuition.

To support the explainability implementation outcomes, it is important to include the dispatchers in the user interface design process. During calls they need to understand and evaluate the explanations very quickly, which makes it of utmost importance that they have input on what should be presented and that the dispatchers are adequately trained and instructed how to correctly interpret the explanations to avoid invalid conclusions in a time-sensitive context. Additionally, it is also important to build their understanding of the complete system, starting from the type of data and data collection method, the AI system's functioning, and the limitations of the explanation it provides [13].

**Designated role in decision-making.**   When looking at the third layer, we consider that explainability can promote the use of the system as originally intended, namely as a CDSS that promotes algorithm-based decision making [75]. To foster critical assessment, instructions should be presented to dispatchers in a way that promotes engagement with the tool and

encourages them to familiarize themselves with its functioning. If offered insights into the inner workings of the system and given the opportunity to provide feedback and improve its performance in an iterative process, dispatchers may find it easier to build trust.

## Discussion

Our findings suggest that omitting explainability will lead to challenges in gaining the users' trust in the model. Adding explainability could foster such trust. It is, however, challenging to ensure the validity and usefulness of explanations to users of the system. This is the case because explanations must be tailored to the specific use case, that may vary in terms of technical aspects, users and the designated role of the system. In contrast to simple models that often provide a direct form of explanation, in a black-box model as given in our use case, the challenge remains to provide explanations that are supported by robust validation.

### Explainability: Is the current debate too theoretical?

A major scientific debate revolves around the question of whether explainability is a needed characteristic for CDSSs. At first glance, it seems as if this debate is dominated by two differing views, where one side argues that a focus on well-validated performance is sufficient [24,29], whereas the other side points out the usefulness of explainability and even argues for normative reasons to adopt explainability [10,13,18,34]. There is, however, an increasingly relevant third perspective that does not question the necessity of explainability but rather points out that the benefits associated with explainability will only be achieved by applying interpretable algorithms. The proponents of this view argue that black-box explanations are based on shaky theoretical assumptions and that such post-hoc approximations of the underlying models will not produce reliable explainability [17,22,28,45].

Based on previous works [13,18,21,22,37] and the results of the current analysis, we argue that if well-validated and methodologically sound explanations that are intuitive and easy to grasp for users were readily available for a black-box system, there would be no valid argument against their inclusion. It seems evident that confronted with a black-box, it is unproportionally difficult for dispatchers to establish trust in a system that is designed to support them in their decision-making, especially if they are—as in the current use-case—likely responsible and legally liable for the final decision. This may result—as in the first validation study of the given system—in ignoring the black box system, or worse by irritating the users, leading to a worse performance. We believe that these considerations apply to CDSSs in general.

There is, however, the possibility that proponents of explainability as a concept overlook technological hurdles to derive explanations for black-box models. It is not straightforward to obtain validated and technologically sound explanations for black-box models, especially for the very popular artificial neural network applications. In this case, it is necessary to carefully weigh the advantages and disadvantages of different techniques for generating explanations, as well as to analyze which of these techniques can be understood by end users [76]. It is also critical that regulators are aware of the benefits and limitations of interpretable and explainable algorithms. Nonetheless, no matter which type of explainability will be applied for a CDSS, we argue that it is critical to include the end users in the design, tailoring the system to their needs, and to instruct them on the used data and algorithms, as well as the system's performance and limitations and informing them on potential biases and limitations in the training data that might influence effectiveness when used on a wider population. These steps are recommended for trustworthy AI [13] and they are, also in our opinion, essential for fostering trust in CDSSs. More importantly, adding explanations should not be an excuse to omit necessary clinical validation. Implementation of explainability is an additional iteration in the

development process, during which it is ensured that the explanations or interpretations are useful and understandable to the end users. It also helps to identify instances where they might not accurately reflect the system's classification process.

Here, it is important to note that the use case described in this paper has only one primary user group who interacts with the system, i.e., the dispatchers. In other words, dispatchers usually won't need to explain their decisions and how they use the system to anyone. However, there are other types of CDSSs used by clinicians to guide decision-making in patient care (e.g., treatment decisions or lifestyle interventions), which will be visible to two or more parties (e.g., patients or interprofessional teams). In such cases, clinicians may need to be prepared to translate and communicate system-generated explanations to diverse patient populations or colleagues in a comprehensive manner.

Taken together, we believe that the current debate around explainability is 'too theoretical'. It is likely that the question, whether explainability is needed or not and whether explainable or interpretable algorithms are to be preferred, cannot be finally answered on a high and theoretical level. Rather, we need to individually analyze implemented systems and their specific needs with regards to technology, displayed information, user groups, and the intended use of the system in decision making to draw conclusions about explainability requirements. Studying use cases, like the one presented here, can draw attention to the multiple, sometimes conflicting requirements and perspectives present in the clinical setting, which may not emerge from a purely theoretically grounded discourse.

Thus, even if overarching plausible arguments exist, such as for explainability as a concept or against black-box explanations, we caution against any absolute dos or don'ts: for example, against legislation that would make explainability a legal requirement or against rules that would impose a 'ban' on certain types of algorithms a priori. Instead, we advocate for changes to the regulatory pathways for AI in healthcare medical devices which are currently geared towards tools that are rarely updated. In our view, the current system with one regulatory process for all types of tools will unlikely suffice to guide manufacturers towards the appropriate amount of explainability. It also does not allow for rapid technical amendments in light of retraining and clinical feedback.

## Explainability dictates the system role

The analysis of our use case revealed a crucial point that has not gained enough attention in the literature so far. Based on our analysis, we argue that some products that are called CDSSs cannot currently be considered to *support* clinical decisions without additional explainability. Given the time-criticality in decision making in our use case, there is an 'epistemic gap' when the dispatchers disagree with a black-box system. They can only decide between following the system or not; they cannot really use the output generated by the system to make an informed decision. This is exacerbated by the fact that the system is proven to be better at detecting cardiac arrests than the dispatchers. Arguably, the outcome for patients might thus be better if the dispatchers were removed altogether from the decision-making process. Given such a situation, it is not fitting to assume that the system *supports* a decision.

In this context, the German Data Ethics Committee made the following distinction of potential AI tools for medicine [77]: a) algorithm-*based* decisions: human decisions that are supported by AI systems, b) algorithm-*driven* decisions: human decisions that are influenced in a way by an AI system that the decision space is greatly reduced by the AI system, and c) algorithm-*determined* decisions: The decisions are made without human interventions.

For the given use case, we can thus conclude that the system without well-validated explainability most closely relates to an *algorithm-driven* tool. This is because without an explanation,

dispatchers are, in fact, unable to make an informed decision whether or not to trust and comply with the system's recommendation. In other words, they only have the choice whether or not to comply and have no information at their disposal to make or justify their decision. The best outcome would probably be achieved if the dispatchers followed the system in the majority of cases, only on the lookout for crass false negatives or positives. This, however, is not how the system was designed or presented to the dispatchers.

In contrast, with implemented—and validated—explainability, dispatchers would have a way to better understand each prediction presented to them. Being able to look for keywords picked up by the system, they could quickly assess how reliable any prediction is to them. In such a case, the system could rather be considered a decision-support system where the dispatcher is using the 'second opinion' of an AI-dispatcher to fine-tune their decision making.

The intriguing point here is that for certain use cases, the question does not seem to be whether a CDSS should have added explainability. But rather whether explainability is added or not will determine whether the system is a clinical decision *support* system at all. Thus, the question to include explainability—in whatever form—might be driven by a design choice: what kind of tool is actually answering the clinical need best. This has important implications for medical product development and should be addressed in future interdisciplinary research.

### The need for a defined terminology in explainability

Our analysis adds to the growing body of literature outlining the complexity of explainability in the field of medicine and specifically CDSSs. It is likely that with the increasing maturity of this field, clear rules will have to be devised for the use of explainability. In analogy to other concepts, some of these rules will be recommendations, some will become scientific and product development standards, and some will be required by law. No matter their level of binding, however, it must be clear what these terms mean. As an example, the current Ethics *guidelines for trustworthy AI* of the European Union refer to "explicability" as an ethical principle of trustworthy AI and transparency as a way of realizing it [13], and while "explicability" is not precisely defined, the document provides sub concepts for increasing transparency, with explainability as one of them. And, as outlined earlier, explainability (and interpretability) are used in the scientific literature with very different meanings. It leads to a status-quo where each publication needs to define in detail what the used terms mean, which in turn causes an inflation of definitions in use.

Next to expectable confusion of all stakeholders involved, such a situation can have dire consequences for patients. In the end, every scientific advance will only find its way to the clinical setting to help patients via a product development path. The current situation with regards to explainability leads to major uncertainties for developers. This can become problematic for companies and start-ups developing products and preparing for regulatory approval. They might apply black-box algorithms with certain types of explanation algorithms to fulfill the need for "explicability" and could, in the future, be confronted with sudden regulatory changes, such as rules to apply interpretable algorithms only.

There is thus a major need to harmonize the terminology. We strongly suggest an international workshop with the creation of an 'explainability white paper' clearly defining all relevant terms to facilitate scientific research, policymaking, and medical device development.

### Conclusion

We conclude that whether explainability can provide added value to CDSSs depends on several key questions: technical feasibility, the level of validation in case of explainable algorithms, the exact characteristics of the context in which the system is implemented (e.g., the time criticality

of the decision making), the designated role in the decision-making process (algorithm-based, -driven, or -determined), and the key user group(s). We deem it likely that the role of explainability cannot be answered definitively at a high, theoretical level. Instead, each system developed for the clinical setting will require an individualized assessment of explainability needs. Thus, this paper cannot provide a universal answer to the question of whether and in what form explainability should be considered a core requirement for AI-powered CDSS. Rather, it illustrates the importance of studying individual real-world applications to uncover areas of concern and to anticipate future ethical and practical challenges. It also highlights the need for an interdisciplinary approach, like the Z-Inspection® process, to study and assess AI-powered CDSS given the vast array of stakeholder affected by these solutions and the interdependencies to be considered. This paper presents an example of what such interdisciplinary assessments for explainability requirements could look like in practice.

## Author Contributions

**Conceptualization:** Julia Amann, Dennis Vetter, Roberto V. Zicari, Vince Istvan Madai.

**Formal analysis:** Julia Amann, Dennis Vetter, Vince Istvan Madai.

**Investigation:** Julia Amann, Dennis Vetter, Stig Nikolaj Blomberg, Helle Collatz Christensen, Megan Coffee, Sara Gerke, Thomas K. Gilbert, Thilo Hagendorff, Sune Holm, Michelle Livne, Andy Spezzatti, Inga Strümke, Roberto V. Zicari, Vince Istvan Madai.

**Methodology:** Julia Amann, Dennis Vetter, Vince Istvan Madai.

**Writing – original draft:** Julia Amann, Dennis Vetter, Vince Istvan Madai.

**Writing – review & editing:** Julia Amann, Dennis Vetter, Stig Nikolaj Blomberg, Helle Collatz Christensen, Megan Coffee, Sara Gerke, Thomas K. Gilbert, Thilo Hagendorff, Sune Holm, Michelle Livne, Andy Spezzatti, Inga Strümke, Roberto V. Zicari, Vince Istvan Madai.

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
