## [Decision Letter · Decision Letter 0]

26 Oct 2021

PDIG-D-21-00082

To explain or not to explain? - A Case Study of Artificial Intelligence Explainability in Clinical Decision Support Systems

PLOS Digital Health

Dear Dr. Amann,

Thank you for submitting your manuscript to PLOS Digital Health. After careful consideration, we feel that it has merit but does not fully meet PLOS Digital Health’s publication criteria as it currently stands. Therefore, we invite you to submit a revised version of the manuscript that addresses the points raised during the review process.

The authors can provide point-by-point replies to all review comments so that the revision can be further improved.

We look forward to receiving your revised manuscript.

Kind regards,

Henry Horng-Shing Lu

Section Editor

PLOS Digital Health

Journal Requirements:

1. Please provide separate figure files in .tif or .eps format only and remove any figures embedded in your manuscript file. Please ensure that all files are under our size limit of 20MB.  

Once you've converted your files to .tif or .eps, please also make sure that your figures meet our format requirements

For more information about how to convert your figure files please see our guidelines: https://journals.plos.org/digitalhealth/s/figures

2. Please provide a complete Data Availability Statement in the submission form, ensuring you include all necessary access information or a reason for why you are unable to make your data freely accessible. Note that it is not acceptable for the authors to be the sole named individuals responsible for ensuring data access.

PLOS defines a study's minimal data set as the underlying data used to reach the conclusions drawn in the manuscript and any additional data required to replicate the reported study findings in their entirety. Any potentially identifying patient information must be fully anonymized. 

If your research concerns only data provided within your submission, please write "All data are in the manuscript and/or supporting information files" as your Data Availability Statement.

3. We do not publish any copyright or trademark symbols that usually accompany proprietary names, eg (R), (C), or TM  (e.g. next to drug or reagent names). Therefore please remove all instances of trademark/copyright symbols throughout the text, including Z-Inspection® on pages 1, 4, 9, 22, and 29.

Additional Editor Comments (if provided):

The authors can provide point-by-point replies to all review comments so that the revision can be further improved.

Reviewers' comments:

Reviewer's Responses to Questions

**Comments to the Author**

1. Does this manuscript meet PLOS Digital Health’s publication criteria? Is the manuscript technically sound, and do the data support the conclusions? The manuscript must describe methodologically and ethically rigorous research with conclusions that are appropriately drawn based on the data presented.

Reviewer #1: Partly

Reviewer #2: Partly

Reviewer #3: Yes

2. Has the statistical analysis been performed appropriately and rigorously?

Reviewer #1: N/A

Reviewer #2: N/A

Reviewer #3: N/A

3. Have the authors made all data underlying the findings in their manuscript fully available (please refer to the Data Availability Statement at the start of the manuscript PDF file)?

Reviewer #1: No

Reviewer #2: No

Reviewer #3: No

4. Is the manuscript presented in an intelligible fashion and written in standard English?

Reviewer #1: Yes

Reviewer #2: Yes

Reviewer #3: Yes

5. Review Comments to the Author

Reviewer #1: The paper presents an ethical review of arguments for and against explainability for AI-powered CDSS in medicine. The authors present findings from a workshop and draw on a qualitative case study to foster the largely theoretical debate about explanations in medical AI. The subject is topical, clinically and scientifically relevant and overall, the paper is well written.

I enjoyed reading the paper, especially the introduction is well written and well structured. The biggest issue I have with the paper becomes clear from the methods section on, where the paper is confused about the approach, goals and status of the specific case. This is not only methodologically problematic, but also a challenge for assessing the value of the paper and the presented analysis: I do -as it currently stands- not have reasons to trust the claims made in the paper. Below, I outline some specific issues related to this overarching issue:

The methodology section lacks information about the followed approach, included data and the analysis of the qualitative data: what information is included in the case study analysis (interviews, observations at case site?) what was provided and discussed at the workshop, who was there, how many participants, how was the data recorded and analyzed, etc.

The results do not read as a case study: it is for example not clear when the authors and when the participants are ‘speaking’, whether there was agreement/disagreement and what from that discussion ended up in the paper.

The suggestions in results and discussion are rather general about AI systems and explanation and do not refer to the specific use case, while the description of the specific system is quite specific. How generalizable are suggestions that were discussed for this specific system?

There are several ways forward: 1) the paper could be rewritten as a more normative paper, then the empirical case should be removed or can be used as an example; 2) The authors clarify the empirical approach and analysis and rewrite the results in such a way that it becomes clear what the participants argued and whether there was (dis)agreement; 3) the authors use the case as a showcase, then the results section should be more specified for this specific case: when and why is explanation needed for this CDSS?

Other issues in the result section:

clarification required: in the paper is described that the AI system has a ‘superior performance’, yet also the high number of false positives is mentioned. Do the authors refer to the sensitivity of the system with superior performance with the superiority? Can the authors clarify whether that an adequate description?

the authors argue that such systems should be able to deal with various accents and dialects, yet do not mention whether that would be a trade-off with for example accuracy? If so, aren’t there alternatives, eg: no recommendation from the system when dialect is not sufficiently understood/training of employees to know the limits of the system?

authors suggest critical assessment of the systems, for which transparency is suggested to be important. Are there alternatives to this (eg: knowing the reliability of such systems), why is transparancy the best solution?

Issues in the discussion and conclusion

‘Taken together, we believe that the current debate around explainability is ‘too theoretical’. It is likely that the question, whether explainability is needed or not and whether explainable or interpretable algorithms are to be preferred, cannot be finally answered on a high and theoretical level.’ This sounds intuitive, but wasn't that the reason to dive into a use-case? What can we learn from this empirical case study that you dived into?

similar point in the conclusion: ‘ Thus, we deem it likely that the role of explainability cannot be answered definitively at a high, theoretical level.’ Seems an unsatisfying conclusion for an empirically-informed paper.

‘ Thus, even if overarching plausible arguments exist, such as for explainability as a concept or against black-box explanations, we caution against any absolute dos or don'ts: for example, against legislation that would make explainability a legal requirement or against rules that would impose a ‘ban’ on certain types of algorithms a priori.’ Could you add a more constructive note: what do you suggest using instead?

‘ For the given use case, we can thus conclude that the system without well-validated explainability most closely relates to an algorithm-driven tool.’ This doesn't follow from the results as they stand and requires more profound argumentation.

433‘ The autonomy of the dispatchers to make decisions is greatly reduced.’ This claim is ot supported by the results as they stand

454 ‘ No matter their level of binding, however, it must be clear what these terms mean. As an example, the current Guidelines for trustworthy AI of the European Union refer to “explicability” as an important concept in making AI transparent (13). It is, however, unclear what this term means, especially as the same document also refers to explainability as a sub-concept.’ Indeed, conceptual clarification is necessary and useful, the following paper presents an attempt to differentiate between transparency and explainability and may be helpful for your analysis: https://jme.bmj.com/content/47/5/329.info

Reviewer #2: Synopsis of my understanding of the article and its arguments.

The paper presents a review of arguments in favor and against explainability for AI-powered Clinical Decision Support Systems (CDSS), employing a qualitative case study and a normative analysis to explore different development scenarios for the CDSS system - with and without explainability. The author's analysis suggests that whether explainability can provide added value to CDSS depends on technical feasibility, the level of validation in case of explainable algorithms, the characteristics of the context in which the system is implemented, the designated role in the decision-making process, and the key user group(s). The authors conclude that each CDSS will require an individualized assessment of explainability needs.

Is the submission original and provocative? Is its argument made in a coherent and succinct manner? Is it suitable for PLOS Digital Health?

The submission is somewhat provocative, arguments are clear, and over-all this article fits the theme of PLOS Digital Health. The article is building upon the theme of trying to explain algorithms in various domains, including healthcare. The reviewer believes this paper addresses an inherently interesting topic: a real-world account of the role of explainability (here, an AI system used to detect cardiac arrest in an emergency call setting in Denmark). The following comments are suggestions for making improvements on an already solid paper.

What are the strengths and weaknesses of the submission?

Strengths:

Thank you for addressing the issue of defining explainability, especially the nuance between explainable and interpretable algorithms.

Thank you for discussing the technical challenges inherent in explainability, including the all-important human (here, the HCP, dispatcher) requirement of trust.

Weaknesses:

My main criticism of the article relates to the assertion in the title: "To explain or not to explain? . . ." Are you arguing that explanations may not be necessary? Or are you arguing that explanations are necessary, but time, user, and context specific. I interpret your paper as pursuing the latter. For example, is explainability, in your assessment, real-time explainability (like in an emergency situation) or can explainability be time-shifted (like during a training)?

Finally, I am having trouble with the last sentence. “Lastly, our paper demonstrates that interdisciplinary assessments like the Z-Inspection® process are invaluable tools to analyse the characteristics and the impact of AI-based CDSSs.” I tend to agree with this, but I’m not sure that your paper is demonstrating this process as “invaluable.”

Could improvements and revisions (minor, major) address the weaknesses identified? If so, please provide a specific listing of such improvements to support such a recommendation.

See above.

Reviewer’s Boilerplate Comments

The title of the article is not totally clear, and the abstract could be more informative.

The text is generally well-written, free of unnecessary jargon, and streamlined.

The significance of the problem in the paper is explained and the significance is compelling.

The ideas are developed logically and thoroughly.

Different viewpoints could be better acknowledged.

Sources are well-integrated into the paper (not for the sake of adding sources).

Word choice could be more specific and concrete (see comment on "To explain or not to explain? . . .")

Sentences are clear.

Overall organization of the argument is effective.

No grammatical errors.

Reviewer #3: The authors seek to find out the role explainability plays in the adaptation/performance of clinical decision support systems. The results are based on a Z-inspection process where some of the authors in this paper developed, and four consecutive workshops assessing the truthworthiness of AI systems, using an AI-based CDSS system to recognize out-of-hospital cardiac arrest from audio files of calls to the emergency medical dispatch center (reference 31). The authors took the outcomes of the workshops and analyzed them focusing on the issue of explainability. The reviewer note that distinguishing the case of employing interpretable algorithms from the case of block-box algorithms amended with an explanation algorithms is very important point. Also the three layers approach to deliberate and present the findings is quite fruitful. The results in this paper are helpful for people working/debating the explainability of AI machine algorithms, instructive for people working on assessing AI algorithms, however, for people who want to understand the process of deriving the results from inputs which is the outcome of reference 31, they do encounter some obstacles. The reviewer would like to suggest:

1. The authors briefly recap reference 31 for readers to get a better pictures

2. Give some quotations from the workshops to support summarized arguments/observations.

6. PLOS authors have the option to publish the peer review history of their article (what does this mean?). If published, this will include your full peer review and any attached files.

**Do you want your identity to be public for this peer review?** For information about this choice, including consent withdrawal, please see our Privacy Policy.

Reviewer #1: No

Reviewer #2: No

Reviewer #3: No

---

## [Decision Letter · Decision Letter 1]

3 Jan 2022

To explain or not to explain? - Artificial Intelligence Explainability in Clinical Decision Support Systems

PDIG-D-21-00082R1

Dear Dr. Amann,

We're pleased to inform you that your manuscript has been judged scientifically suitable for publication and will be formally accepted for publication once it meets all outstanding technical requirements. 

Within one week, you'll receive an e-mail detailing the required amendments. When these have been addressed, you'll receive a formal acceptance letter and your manuscript will be scheduled for publication. The journal will begin publishing content in early 2022.

An invoice for payment will follow shortly after the formal acceptance. To ensure an efficient process, please log into Editorial Manager at https://www.editorialmanager.com/pdig/ click the 'Update My Information' link at the top of the page, and double check that your user information is up-to-date. If you have any billing related questions, please contact our Author Billing department directly at authorbilling@plos.org.

Kind regards,

Henry Horng-Shing Lu

Section Editor

PLOS Digital Health

Additional Editor Comments (optional):

Reviewers' comments:

Reviewer's Responses to Questions

**Comments to the Author**

1. If the authors have adequately addressed your comments raised in a previous round of review and you feel that this manuscript is now acceptable for publication, you may indicate that here to bypass the “Comments to the Author” section, enter your conflict of interest statement in the “Confidential to Editor” section, and submit your "Accept" recommendation.

Reviewer #2: All comments have been addressed

Reviewer #3: All comments have been addressed

2. Does this manuscript meet PLOS Digital Health’s publication criteria? Is the manuscript technically sound, and do the data support the conclusions? The manuscript must describe methodologically and ethically rigorous research with conclusions that are appropriately drawn based on the data presented.

Reviewer #2: Partly

Reviewer #3: Yes

3. Has the statistical analysis been performed appropriately and rigorously?

Reviewer #2: N/A

Reviewer #3: Yes

4. Have the authors made all data underlying the findings in their manuscript fully available (please refer to the Data Availability Statement at the start of the manuscript PDF file)?

Reviewer #2: No

Reviewer #3: Yes

5. Is the manuscript presented in an intelligible fashion and written in standard English?

Reviewer #2: Yes

Reviewer #3: Yes

6. Review Comments to the Author

Reviewer #2: N/A

Reviewer #3: no

7. PLOS authors have the option to publish the peer review history of their article (what does this mean?). If published, this will include your full peer review and any attached files.

**Do you want your identity to be public for this peer review?** For information about this choice, including consent withdrawal, please see our Privacy Policy.

Reviewer #2: No

Reviewer #3: No
